# Tree of Thoughts: Deliberate Problem Solving with Large Language Models

**Shunyu Yao**
Princeton University

**Dian Yu**
Google DeepMind

**Jeffrey Zhao**
Google DeepMind

**Izhak Shafran**
Google DeepMind

**Thomas L. Griffiths**
Princeton University

**Yuan Cao**
Google DeepMind

**Karthik Narasimhan**
Princeton University

## Abstract

Language models are increasingly being deployed for general problem solving across a wide range of tasks, but are still confined to token-level, left-to-right decision-making processes during inference. This means they can fall short in tasks that require exploration, strategic lookahead, or where initial decisions play a pivotal role. To surmount these challenges, we introduce a new framework for language model inference, "Tree of Thoughts" (ToT), which generalizes over the popular "Chain of Thought" approach to prompting language models, and enables exploration over coherent units of text ("thoughts") that serve as intermediate steps toward problem solving. ToT allows LMs to perform deliberate decision making by considering multiple different reasoning paths and self-evaluating choices to decide the next course of action, as well as looking ahead or backtracking when necessary to make global choices. Our experiments show that ToT significantly enhances language models' problem-solving abilities on three novel tasks requiring non-trivial planning or search: Game of 24, Creative Writing, and Mini Crosswords. For instance, in Game of 24, while GPT-4 with chain-of-thought prompting only solved 4% of tasks, our method achieved a success rate of 74%. Code repo with all prompts: `https://github.com/princeton-nlp/tree-of-thought-llm`.

## 1 Introduction

Originally designed to generate text, scaled-up versions of language models (LMs) such as GPT [25, 26, 1, 23] and PaLM [5] have been shown to be increasingly capable of performing an ever wider range of tasks requiring mathematical, symbolic, commonsense, and knowledge reasoning. It is perhaps surprising that underlying all this progress is still the original autoregressive mechanism for generating text, which makes token-level decisions one by one and in a left-to-right fashion. Is such a simple mechanism sufficient for a LM to be built toward a general problem solver? If not, what problems would challenge the current paradigm, and what should be alternative mechanisms?

The literature on human cognition provides some clues to answer these questions. Research on "dual process" models suggests that people have two modes in which they engage with decisions – a fast, automatic, unconscious mode ("System 1") and a slow, deliberate, conscious mode ("System 2") [30, 31, 16, 15]. These two modes have previously been connected to a variety of mathematical models used in machine learning. For example, research on reinforcement learning in humans and other animals has explored the circumstances under which they engage in associative "model free" learning or more deliberative "model based" planning [7]. The simple associative token-level choices of LMs are also reminiscent of "System 1", and thus might benefit from augmentation by a more deliberate "System 2" planning process that (1) maintains and explores diverse alternatives for current

37th Conference on Neural Information Processing Systems (NeurIPS 2023).

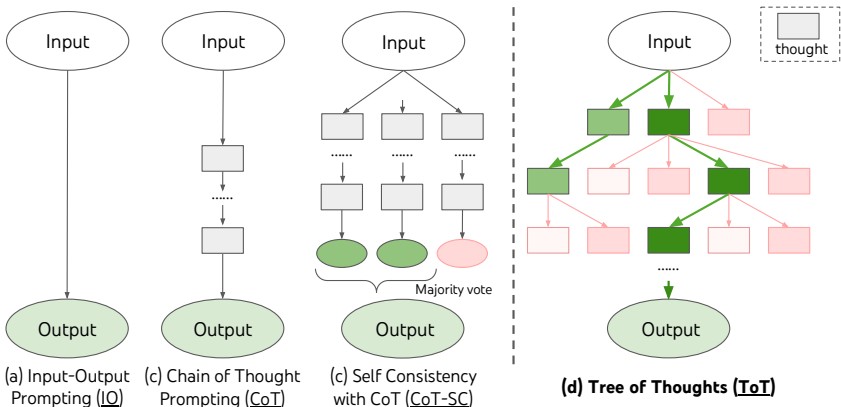

Figure 1: Schematic illustrating various approaches to problem solving with LLMs. Each rectangle box represents a *thought*, which is a coherent language sequence that serves as an intermediate step toward problem solving. See concrete examples of how thoughts are generated, evaluated, and searched in Figures 2,4,6.

choices instead of just picking one, and (2) evaluates its current status and actively looks ahead or backtracks to make more global decisions.

To design such a planning process, we return to the origins of artificial intelligence (and cognitive science), drawing inspiration from the planning processes explored by Newell, Shaw, and Simon starting in the 1950s [21, 22]. Newell and colleagues characterized problem solving [21] as search through a combinatorial problem space, represented as a tree. We thus propose the Tree of Thoughts (ToT) framework for general problem solving with language models. As Figure 1 illustrates, while existing methods (detailed below) sample continuous language sequences for problem solving, ToT actively maintains a tree of thoughts, where each *thought* is a coherent language sequence that serves as an intermediate step toward problem solving (Table 1). Such a high-level semantic unit allows the LM to self-evaluate the progress different intermediate thoughts make towards solving the problem through a deliberate reasoning process that is also instantiated in language (Figures 2,4,6). This implementation of search heuristics via LM self-evaluation and deliberation is novel, as previous search heuristics are either programmed or learned. Finally, we combine this language-based capability to generate and evaluate diverse thoughts with search algorithms, such as breadth-first search (BFS) or depth-first search (DFS), which allow systematic exploration of the tree of thoughts with lookahead and backtracking.

Empirically, we propose three new problems that challenge existing LM inference methods even with the state-of-the-art language model, GPT-4 [23]: Game of 24, Creative Writing, and Crosswords (Table 1). These tasks require deductive, mathematical, commonsense, lexical reasoning abilities, and a way to incorporate systematic planning or search. We show ToT obtains superior results on all three tasks by being general and flexible enough to support different levels of thoughts, different ways to generate and evaluate thoughts, and different search algorithms that adapt to the nature of different problems. We also analyze how such choices affect model performances via systematic ablations and discuss future directions to better train and use LMs.

## 2   Background

We first formalize some existing methods that use large language models for problem-solving, which our approach is inspired by and later compared with. We use $p_\theta$ to denote a pre-trained LM with parameters $\theta$, and **lowercase letters** $x, y, z, s, \cdots$ **to denote a language sequence**, i.e. $x = (x[1], \cdots, x[n])$ where each $x[i]$ is a token, so that $p_\theta(x) = \prod_{i=1}^{n} p_\theta(x[i]|x[1...i])$. We use uppercase letters $S, \cdots$ to denote a collection of language sequences.

**Input-output (IO) prompting** is the most common way to turn a problem input $x$ into output $y$ with LM: $y \sim p_\theta(y|\text{prompt}_{IO}(x))$, where $\text{prompt}_{IO}(x)$ wraps input $x$ with task instructions and/or few-shot input-output examples. For simplicity, let us denote $p_\theta^{\text{prompt}}(\text{output} \mid \text{input}) = p_\theta(\text{output} \mid \text{prompt}(\text{input}))$, so that IO prompting can be formulated as $y \sim p_\theta^{IO}(y|x)$.

**Chain-of-thought (CoT) prompting** [38] was proposed to address cases where the mapping of input $x$ to output $y$ is non-trivial (e.g. when $x$ is a math question and $y$ is the final numerical answer). The key idea is to introduce a chain of *thoughts* $z_1, \cdots, z_n$ to bridge $x$ and $y$, where each $z_i$ is a coherent language sequence that serves as a meaningful intermediate step toward problem solving (e.g. $z_i$ could be an intermediate equation for math QA). To solve problems with CoT, each thought $z_i \sim p_\theta^{CoT}(z_i \mid x, z_{1 \cdots i-1})$ is sampled sequentially, then the output $y \sim p_\theta^{CoT}(y|x, z_{1 \cdots n})$. In practice, $[z_{1 \cdots n}, y] \sim p_\theta^{CoT}(z_{1 \cdots n}, y|x)$ is sampled as a continuous language sequence, and the **decomposition** of thoughts (e.g. is each $z_i$ a phrase, a sentence, or a paragraph) is left ambiguous.

**Self-consistency with CoT (CoT-SC)** [36] is an ensemble approach that samples $k$ i.i.d. chains of thought: $[z_{1 \cdots n}^{(i)}, y^{(i)}] \sim p_\theta^{CoT}(z_{1 \cdots n}, y|x)$ $(i = 1 \cdots k)$, then returns the most frequent output: $\arg\max_y \#\{i \mid y^{(i)} = y\}$. CoT-SC improves upon CoT, because there are generally different thought processes for the same problem (e.g. different ways to prove the same theorem), and the output decision can be more faithful by exploring a richer set of thoughts. However, within each chain there is no local exploration of different thought steps, and the "most frequent" heuristic only applies when the output space is limited (e.g. multi-choice QA).

## 3 Tree of Thoughts: Deliberate Problem Solving with LM

> *A genuine problem-solving process involves the repeated use of available informa-*
> *tion to initiate exploration, which discloses, in turn, more information until a way*
> *to attain the solution is finally discovered.*—— *Newell et al. [21]*

Research on human problem-solving suggests that people search through a combinatorial problem-space – a tree where the nodes represent partial solutions, and the branches correspond to operators that modify them [21, 22]. Which branch to take is determined by heuristics that help to navigate the problem-space and guide the problem-solver towards a solution. This perspective highlights two key shortcomings of existing approaches that use LMs to solve general problems: 1) Locally, they do not explore *different* continuations within a thought process – the branches of the tree. 2) Globally, they do not incorporate any type of planning, lookahead, or backtracking to help evaluate these different options – the kind of heuristic-guided search that seems characteristic of human problem-solving.

To address these shortcomings, we introduce *Tree of Thoughts (ToT)*, a paradigm that allows LMs to explore multiple reasoning paths over thoughts (Figure 1(c)). ToT frames any problem as a search over a tree, where each node is a **state** $s = [x, z_{1 \cdots i}]$ representing a partial solution with the input and the sequence of thoughts so far. A specific instantiation of ToT involves answering four questions: 1. How to **decompose** the intermediate process into thought steps; 2. How to **generate** potential thoughts from each state; 3. How to heuristically **evaluate** states; 4. What **search** algorithm to use.

**1. Thought decomposition.** While CoT samples thoughts coherently without explicit decomposition, ToT leverages problem properties to design and decompose intermediate thought steps. As Table 1 shows, depending on different problems, a thought could be a couple of words (Crosswords), a line of equation (Game of 24), or a whole paragraph of writing plan (Creative Writing). In general, a thought should be "small" enough so that LMs can generate promising and diverse samples (e.g. generating a whole book is usually too "big" to be coherent), yet "big" enough so that LMs can evaluate its prospect toward problem solving (e.g. generating one token is usually too "small" to evaluate).

**2. Thought generator** $G(p_\theta, s, k)$. Given a tree state $s = [x, z_{1 \cdots i}]$, we consider two strategies to generate $k$ candidates for the next thought step:

  (a) **Sample** i.i.d. thoughts from a CoT prompt (Creative Writing, Figure 4): $z^{(j)} \sim p_\theta^{CoT}(z_{i+1}|s) = p_\theta^{CoT}(z_{i+1}|x, z_{1 \cdots i})$ $(j = 1 \cdots k)$. This works better when the thought space is rich (e.g. each thought is a paragraph), and i.i.d. samples lead to diversity;

  (b) **Propose** thoughts sequentially using a "propose prompt" (Game of 24, Figure 2; Crosswords, Figure 6): $[z^{(1)}, \cdots, z^{(k)}] \sim p_\theta^{propose}(z_{i+1}^{(1 \cdots k)} \mid s)$. This works better when the thought space is more constrained (e.g. each thought is just a word or a line), so proposing different thoughts in the same context avoids duplication.

**3. State evaluator** $V(p_\theta, S)$. Given a frontier of different states, the state evaluator evaluates the progress they make towards solving the problem, serving as a *heuristic* for the search algorithm to determine which states to keep exploring and in which order. While heuristics are a standard approach to solving search problems, they are typically either programmed (e.g. DeepBlue [3]) or

learned (e.g. AlphaGo [29]). We propose a third alternative, by using the LM to deliberately reason about states. When applicable, such a deliberate heuristic can be more flexible than programmed rules, and more sample-efficient than learned models. Similar to the thought generator, we consider two strategies to evaluate states either independently or together:

(a) **Value** each state independently: $V(p_\theta, S)(s) \sim p_\theta^{value}(v|s) \ \forall s \in S$, where a value prompt reasons about the state $s$ to generate a scalar value $v$ (e.g. 1-10) or a classification (e.g. sure/likely/impossible) that could be heuristically turned into a value. The basis of such evaluative reasoning can vary across problems and thought steps. In this work, we explore evaluation via few *lookahead* simulations (e.g. quickly confirm that 5, 5, 14 can reach 24 via 5 + 5 + 14, or "hot_l" can mean "inn" via filling "e" in "_") plus commonsense (e.g. 1 2 3 are too small to reach 24, or no word can start with "tzxc"). While the former might promote "good" states, the latter could help eliminate "bad" states. Such valuations do not need to be perfect, and only need to be approximately helpful for decision making.

(b) **Vote** across states: $V(p_\theta, S)(s) = \mathbb{1}[s = s^*]$, where a "good" state $s^* \sim p_\theta^{vote}(s^*|S)$ is voted out based on deliberately comparing different states in $S$ in a vote prompt. When problem success is harder to directly value (e.g. passage coherency), it is natural to to instead compare different partial solutions and vote for the most promising one. This is similar in spirit to a "step-wise" self-consistency strategy, i.e. cast "which state to explore" as a multi-choice QA, and use LM samples to vote for it.

For both strategies, we could prompt the LM multiple times to aggregate the value or vote results to trade time/resource/cost for more faithful/robust heuristics.

| **Algorithm 1** ToT-BFS($x, p_\theta, G, k, V, T, b$) | **Algorithm 2** ToT-DFS($s, t, p_\theta, G, k, V, T, v_{th}$) |
|---|---|
| **Require:** Input $x$, LM $p_\theta$, thought generator $G()$ & size limit $k$, states evaluator $V()$, step limit $T$, breadth limit $b$. | **Require:** Current state $s$, step $t$, LM $p_\theta$, thought generator $G()$ and size limit $k$, states evaluator $V()$, step limit $T$, threshold $v_{th}$ |
| $S_0 \leftarrow \{x\}$ | **if** $t > T$ **then** record output $G(p_\theta, s, 1)$ |
| **for** $t = 1, \cdots, T$ **do** | **end if** |
| $\quad S'_t \leftarrow \{[s, z] \mid s \in S_{t-1}, z_t \in G(p_\theta, s, k)\}$ | **for** $s' \in G(p_\theta, s, k)$ **do** ▷ sorted candidates |
| $\quad V_t \leftarrow V(p_\theta, S'_t)$ | $\quad$ **if** $V(p_\theta, \{s'\})(s) > v_{thres}$ **then** ▷ pruning |
| $\quad S_t \leftarrow \arg\max_{S \subset S'_t, \|S\|=b} \sum_{s \in S} V_t(s)$ | $\quad\quad$ DFS($s', t+1$) |
| **end for** | $\quad$ **end if** |
| **return** $G(p_\theta, \arg\max_{s \in S_T} V_T(s), 1)$ | **end for** |

**4. Search algorithm.** Finally, within the ToT framework, one can plug and play different search algorithms depending on the tree structure. We explore two relatively simple search algorithms and leave more advanced ones (e.g. A* [11], MCTS [2]) for future work:

(a) **Breadth-first search (BFS)** (Algorithm 1) maintains a set of the $b$ most promising states per step. This is used for Game of 24 and Creative Writing where the tree depth is limit ($T \leq 3$), and initial thought steps can be evaluated and pruned to a small set ($b \leq 5$).

(b) **Depth-first search (DFS)** (Algorithm 2) explores the most promising state first, until the final output is reached ($t > T$), or the state evaluator deems it impossible to solve the problem from the current $s$ ($V(p_\theta, \{s\})(s) \leq v_{th}$ for a value threshold $v_{th}$). In the latter case, the subtree from $s$ is *pruned* to trade exploration for exploitation. In both cases, DFS *backtracks* to the parent state of $s$ to continue exploration.

Conceptually, ToT has several benefits as a method for general problem-solving with LMs: (1) *Generality*. IO, CoT, CoT-SC, and self-refinement can be seen as special cases of ToT (i.e. trees of limited depth and breadth; Figure 1). (2) *Modularity*. The base LM, as well as the thought decomposition, generation, evaluation, and search procedures can all be varied independently. (3) *Adaptability*. Different problem properties, LM capabilities, and resource constraints can be accommodated. (4) *Convenience*. No extra training is needed, just a pre-trained LM is sufficient. The next section will show how these conceptual benefits translate to strong empirical performance in different problems.

## 4 Experiments

We propose three tasks that are hard even when sampling from the state-of-the-art language model, GPT-4 [23], using standard IO prompting or chain-of-thought (CoT) prompting. We show how

|  | Game of 24 | Creative Writing | 5x5 Crosswords |
| --- | --- | --- | --- |
| **Input** | 4 numbers (4 9 10 13) | 4 random sentences | 10 clues (h1. presented;..) |
| **Output** | An equation to reach 24 (13-9)*(10-4)=24 | A passage of 4 paragraphs ending in the 4 sentences | 5x5 letters: SHOWN; WIRRA; AVAIL; ... |
| **Thoughts** | 3 intermediate equations (13-9=4 (left 4,4,10); 10-4=6 (left 4,6); 4*6=24) | A short writing plan (1. Introduce a book that connects...) | Words to fill in for clues: (h1. shown; v5. naled; ...) |
| **#ToT steps** | 3 | 1 | 5-10 (variable) |

Table 1: Task overview. Input, output, thought examples are in blue.

deliberate search in trees of thoughts (ToT) produces better results, and more importantly, interesting and promising new ways to use language models to solve problems requiring search or planning. Unless otherwise stated, we perform experiments using a Chat Completion mode GPT-4[1] with a sampling temperature of 0.7.

## 4.1 Game of 24

Game of 24 is a mathematical reasoning challenge, where the goal is to use 4 numbers and basic arithmetic operations (+-*/) to obtain 24. For example, given input "4 9 10 13", a solution output could be "(10 - 4) * (13 - 9) = 24".

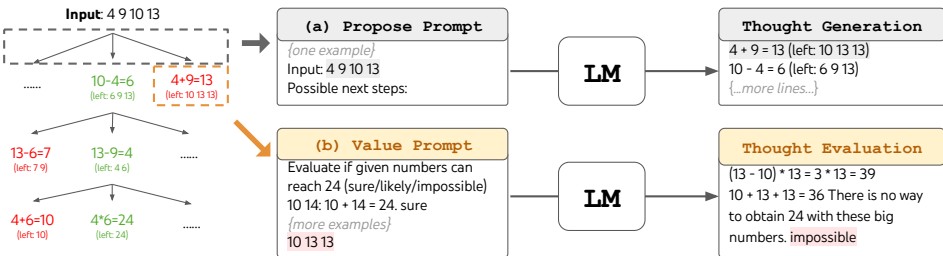

Figure 2: ToT in a game of 24. The LM is prompted for (a) thought generation and (b) valuation.

**Task Setup.** We scrape data from 4nums.com, which has 1,362 games that are sorted from easy to hard by human solving time, and use a subset of relatively hard games indexed 901-1,000 for testing. For each task, we consider the output as success if it is a valid equation that equals 24 and uses the input numbers each exactly once. We report the success rate across 100 games as the metric.

**Baselines.** We use a standard input-output (IO) prompt with 5 in-context examples. For chain-of-thought (CoT) prompting, we augment each input-output pair with 3 intermediate equations, each operating on two remaining numbers. For example, given input "4 9 10 13", the thoughts could be "13 - 9 = 4 (left: 4 4 10); 10 - 4 = 6 (left: 4 6); 4 * 6 = 24 (left: 24)". For each game, we sample IO and CoT prompting for 100 times for average performance. We also consider a CoT self-consistency baseline, which takes the majority output from 100 CoT samples, and an iterative-refine approach on top of an IO sample for at most 10 iterations. At each iteration, the LM is conditioned on all previous history to "reflect on your mistakes and generate a refined answer" if the output is incorrect. Note that it uses groundtruth feedback signals about equation correctness.

**ToT Setup.** To frame Game of 24 into ToT, it is natural to decompose the thoughts into 3 steps, each an intermediate equation. As shown in Figure 2(a), at each tree node, we exact the remaining numbers and prompt the LM to propose some possible next steps. The same "propose prompt" is used for all 3 thought steps, though it only has one example with 4 input numbers. We perform a breadth-first search (BFS) in ToT, where at each step we keep the best $b = 5$ candidates. To perform deliberate BFS in ToT, as shown in Figure 2(b), we prompt LM to evaluate each thought candidate as "sure/maybe/impossible" with regard to reaching 24. The aim is to promote correct partial solutions that can be verdicted within few lookahead trials, and eliminate impossible partial solutions based on "too big/small" commonsense, and keep the rest "maybe". We sample values 3 times for each thought.

[1]Experiments were done between May 5-16, 2023.

| Method | Success |
|--------|---------|
| IO prompt | 7.3% |
| CoT prompt | 4.0% |
| CoT-SC (k=100) | 9.0% |
| ToT (ours) (b=1) | 45% |
| ToT (ours) (b=5) | **74%** |
| IO + Refine (k=10) | 27% |
| IO (best of 100) | 33% |
| CoT (best of 100) | 49% |

Table 2: Game of 24 Results.

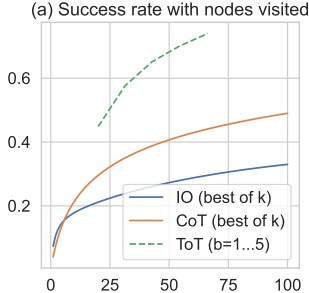
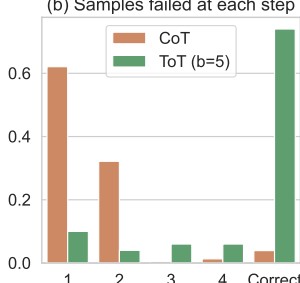

Figure 3: Game of 24 (a) scale analysis & (b) error analysis.

**Results.** As shown in Table 2, IO, CoT, and CoT-SC prompting methods perform badly on the task, achieving only 7.3%, 4.0%, and 9.0% success rates. In contrast, ToT with a breadth of $b = 1$ already achieves a success rate of $45\%$, while $b = 5$ achieves $74\%$. We also consider an oracle setup for IO/CoT, by calculating the success rate using best of $k$ samples ($1 \leq k \leq 100$). To compare IO/CoT (best of k) with ToT, we consider calculating the tree nodes visited per task in ToT across $b = 1 \cdots 5$, and map the 5 success rates in Figure 3(a), treating IO/CoT (best of $k$) as visiting $k$ nodes in a bandit. Not surprisingly, CoT scales better than IO, and best of 100 CoT samples achieve a success rate of $49\%$, but still much worse than exploring more nodes in ToT ($b > 1$).

**Error analysis.** Figure 3(b) breaks down at which step CoT and ToT samples fail the task, i.e. the thought (in CoT) or all $b$ thoughts (in ToT) are invalid or impossible to reach 24. Notably, around 60% of CoT samples already failed the task after generating the first step, or equivalently, the first three words (e.g. "$4 + 9$"). This highlights the issues with direct left-to-right decoding.

## 4.2 Creative writing

Next, we invent a creative writing task where the input is 4 random sentences and the output should be a coherent passage with 4 paragraphs that end in the 4 input sentences respectively. Such a task is open-ended and exploratory, and challenges creative thinking as well as high-level planning.

**Task setup.** We sample random sentences from randomwordgenerator.com to form 100 inputs, and there is no groundtruth passage for each input constraint. As we find that GPT-4 can follow the input constraints most of the time, we focus on evaluating passage coherency in two ways: using a GPT-4 zero-shot prompt to provide a 1-10 scalar score, or using human judgments to compare pairs of outputs from different methods. For the former, we sample 5 scores and average them for each task output, and we find these 5 scores usually consistent, with a standard deviation of around $0.56$ on average across outputs. For the latter, we employ a subset of the authors in a blind study to compare the coherency of CoT vs. ToT generated passage pairs, where the order of passages is random flipped over 100 inputs.

**Baselines.** Given the creative nature of the task, both IO and CoT prompts are zero-shot. While the former prompts the LM to directly generate a coherent passage given input constraints, the latter prompts the LM to first make a brief plan then write the passage, i.e. the plan serves as the intermediate thought step. We generate 10 IO and CoT samples per task. We also consider an iterative-refine ($k \leq 5$) method on top of a random IO sample for each task, where the LM is conditioned on input constraints and the last generated passage to decide if the passage is already "perfectly coherent", and if not generate a refined one.

**ToT setup.** We build a ToT with depth 2 (and only 1 intermediate thought step) — the LM first generates $k = 5$ plans and votes for the best one (Figure 4), then similarly generate $k = 5$ passages based on the best plan then vote for the best one. Here the breadth limit $b = 1$, as only one choice is kept per step. A simple zero-shot vote prompt ("analyze choices below, then conclude which is most promising for the instruction") is used to sample 5 votes at both steps.

**Results.** Figure 5(a) shows average GPT-4 scores across 100 tasks, where ToT (7.56) is deemed to generate more coherent passages than IO (6.19) and CoT (6.93) on average. While such an automatic metric might be noisy, Figure 5(b) confirms the finding by showing that humans prefer ToT over CoT in 41 out of 100 passage pairs, while only prefer CoT over ToT in 21 (other 38 pairs are found "similarly coherent"). Lastly, iterative-refine is more effective on this natural language task, where

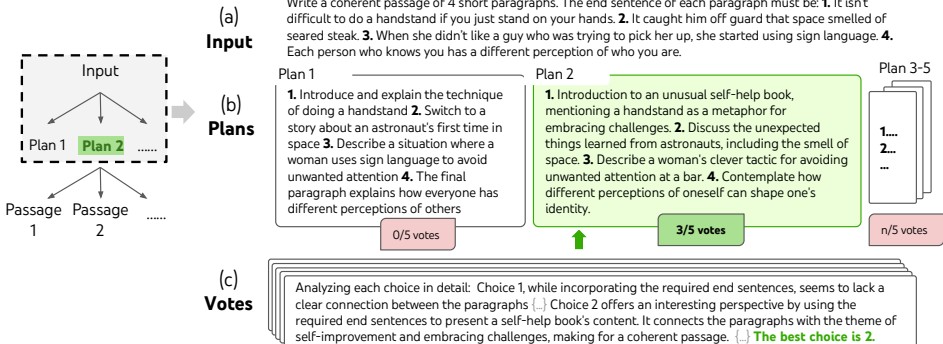

(a) **Input**

Write a coherent passage of 4 short paragraphs. The end sentence of each paragraph must be: **1.** It isn't difficult to do a handstand if you just stand on your hands. **2.** It caught him off guard that space smelled of seared steak. **3.** When she didn't like a guy who was trying to pick her up, she started using sign language. **4.** Each person who knows you has a different perception of who you are.

(b) **Plans**

**Plan 1**

**1.** Introduce and explain the technique of doing a handstand **2.** Switch to a story about an astronaut's first time in space **3.** Describe a situation where a woman uses sign language to avoid unwanted attention **4.** The final paragraph explains how everyone has different perceptions of others

0/5 votes

**Plan 2**

**1.** Introduction to an unusual self-help book, mentioning a handstand as a metaphor for embracing challenges. **2.** Discuss the unexpected things learned from astronauts, including the smell of space. **3.** Describe a woman's clever tactic for avoiding unwanted attention at a bar. **4.** Contemplate how different perceptions of oneself can shape one's identity.

3/5 votes

**Plan 3-5**

1....
2...
...

n/5 votes

(c) **Votes**

Analyzing each choice in detail: Choice 1, while incorporating the required end sentences, seems to lack a clear connection between the paragraphs [...] Choice 2 offers an interesting perspective by using the required end sentences to present a self-help book's content. It connects the paragraphs with the theme of self-improvement and embracing challenges, making for a coherent passage. [...] **The best choice is 2.**

Figure 4: A step of deliberate search in a randomly picked Creative Writing task. Given the input, the LM samples 5 different plans, then votes 5 times to decide which plan is best. The majority choice is used to consequently write the output passage with the same sample-vote procedure.

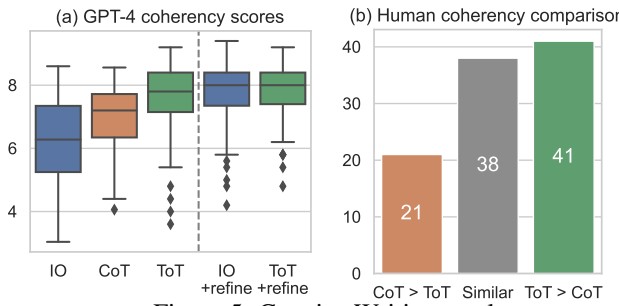

Figure 5: Creative Writing results.

Table 3: Mini Crosswords results.

| Method | Success Rate (%) | | |
|---|---|---|---|
| | Letter | Word | Game |
| IO | 38.7 | 14 | 0 |
| CoT | 40.6 | 15.6 | 1 |
| ToT (ours) | **78** | **60** | **20** |
| +best state | 82.4 | 67.5 | 35 |
| -prune | 65.4 | 41.5 | 5 |
| -backtrack | 54.6 | 20 | 5 |

it improves IO coherency score from 6.19 to 7.67, and ToT coherency score from 7.56 to 7.91. We believe it could be thought of as a third approach to thought generation in the ToT framework, where new thoughts can arise from refining old thoughts instead of i.i.d. or sequentially generated.

## 4.3 Mini crosswords

In Game of 24 and Creative Writing, ToT is relatively shallow — at most 3 thought steps are needed to reach the final output. Here we explore $5 \times 5$ mini crosswords as a harder search problem involving natural language. Again, the goal is not just to solve the task, as more general crosswords can be readily solved with specialized NLP pipelines [34] that leverages large-scale retrieval instead of LM. Rather, we aim to explore the limit of LM as a general problem solver that explores its own thoughts and guides its own exploration with deliberate reasoning as heuristics.

**Task setup.** We scrape data from GooBix, which contains 156 games of $5 \times 5$ mini crosswords. As we observe adjacent games contain similar clues, we use 20 games with indices $1, 6, \cdots, 91, 96$ for testing, and games $136, 141, 146, 151, 156$ for prompting. For each task, the input describes the 5 horizontal clues and 5 vertical clues, and the output should be a board of $5 \times 5 = 25$ letters to solve the crosswords. For evaluation, we consider three levels of success: the portion of correct letters (25 per game), words (10 per game), and games.

**Baselines.** We provide 5 example input-output pairs in the IO prompt, and in the CoT prompt additionally include intermediate words in the order h1..5 then v1..5. We run each prompt for 10 samples and average the results.

**ToT setup.** We leverage a depth-first search (Algorithm 2) that keeps exploring the most promising subsequent word clue until the state is no longer promising, then backtrack to the parent state to explore alternative thoughts. To make search tractable, subsequent thoughts are constrained not to change any filled words or letters, so that the ToT has at most 10 intermediate steps. For thought generation, at each state we translate all existing thoughts (e.g. "h2.motor; h1.tasks" for the state in Figure 6(a)) into letter constraints for remaining clues (e.g. "v1.To heap: tm___;...") and prompt a proposal prompt 5 times to come up with candidates for where and what to fill in the next word. Importantly, we also prompt the LM to give a confidence level for different thoughts, and aggregate

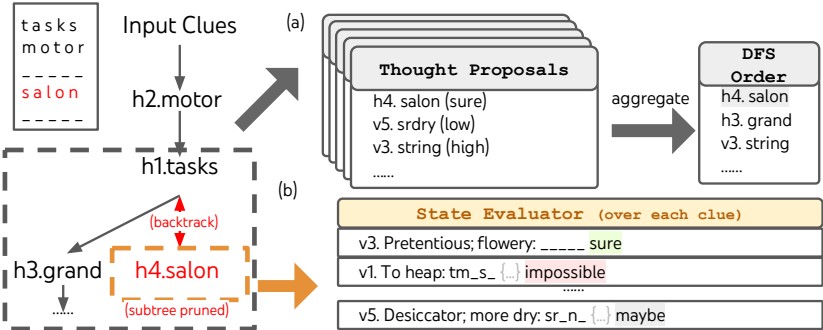

Figure 6: In Mini Crosswords, (a) how thoughts are proposed and aggregated in a priority queue for depth-first search (DFS), and (b) how a state is evaluated based on the possibility of filling in each remaining word clue, and pruned if any remaining clue is deemed not possible to fill by the LM. Then DFS backtracks to the parent state and explore the next promising thought for clue.

these across proposals to obtain a sorted list of next thoughts to explore (Figure 6(a)). For state evaluations, we similarly translate each state into letter constraints for remaining clues, then evaluate for each clue if it is possible to fill given the constraints. If any remaining clue is deemed "impossible" to fill in (e.g. "v1. To heap: tm_s_"), then the exploration of the state's subtree is pruned and DFS backtracks to its parent to explore the next promising thought. We limit DFS search steps to 100, and simply render the deepest explored state (the first explored one if multiple) into the final output.

**Results.** As shown in Table 3, IO and CoT prompting methods perform poorly with a word-level success rate less than $16\%$, while ToT significantly improves all metrics, achieving a word-level success rate of $60\%$ and solving 4 out of 20 games. Such an improvement is not surprising, given IO and CoT lack mechanisms to try different clues, make changes to decisions, or backtrack.

**Oracle and ablation studies.** When outputting from the oracle best DFS state (instead of the heuristically determined best state) per task, ToT performance is even higher and actually solves 7/20 games (Table 3, "+best state"), indicating our simple output heuristics can be readily improved. Interestingly, sometimes when the crosswords game is actually solved, the state evaluator might still deem some words as "impossible" and prune — possibly because $5 \times 5$ crosswords by design have some rare or obselete words that GPT-4 cannot recognize[2]. Given the state evaluation as a pruning heuristic is imperfect, we also explore ablating the pruning, and find the performance generally worse (Table 3, "-prune"). However, it could actually find the correct solution for 4/20 games (though only outputting 1 via heuristic), 3 of which are games ToT+pruning cannot solve within 100 steps. Thus, better heuristics for DFS pruning are critical for problem solving in this case. Lastly, we confirm the importance of backtracking by running an ablation that keeps filling the most promising clue for at most 20 steps, allowing overwrites. This is similar to a "greedy" BFS search with breadth limit of $b = 1$, and performs poorly with a word level success of only $20\%$ (Table 3, "-backtrack").

## 5 Related Work

**Planning and decision making.** Smart planning and decision making are critical to achieving predefined goals. As they are trained on vast amount of world knowledge and human examples, LMs are known to have already absorbed rich commonsense that makes it possible to propose reasonable plans conditioned on problem setting and environmental states [12, 42, 37, 13, 35, 41, 40]. Our proposed ToT approach extends existing planning formulations by considering multiple potentially feasible plans simultaneously at each problem-solving step, and proceeding with the most promising ones. The integration between thought sampling and value feedback organically integrates planning and decision-making mechanisms, enabling effective search inside a solution tree. On the other hand, traditional decision-making procedures usually require training dedicated reward and policy models as in reinforcement learning (for example CHAI [33]), whereas we use the LM itself to provide the value estimates for decision making. RAP [9] is a concurrent work that treats language model

---

[2]For example, "agend" is an obsolete form of "agendum", but GPT-4 deems it a typo for "agenda". External retrieval or web interaction could augment LM for problem solving under knowledge uncertainty.

reasoning as planning with its internal world model, and proposes a MCTS-based method similar to ToT. However, its tasks are simpler than ours, and its framework lacks the modularity to incorporate different tree search algorithms.

**Self-reflection.** Using LLMs to assess the viability of their own predictions is becoming an increasingly important procedure in problem solving. [28, 20, 24] introduced the "self-reflection" mechanism, in which LMs provide feedback to their generation candidates. [4] improves LMs code generation accuracy by injecting feedback messages generated by the LM itself based on its code execution results. Similarly, [17] also introduces "critic" or review steps over the actions and states, deciding the next action to take in solving computer operation tasks. Another recent work very relevant to ours is "self-eval guided decoding" [39]. Similar to our method, self-eval decoding also follows a tree-search procedure with leaves sampled from stochastic beam search decoding, which are then evaluated by LLM itself with carefully prepared self-eval prompts. Their approach however, uses the PAL formulation [8] which represents thoughts as codes, which makes it difficult to tackle challenging tasks like creative writing which we consider in this paper. Our Tree-of-Thought formulation is thus more versatile and handles challenging tasks on which GPT-4 only achieves very low accuracy with standard prompts.

**Program-guided LLM generation.** Our proposal is also related to recent advancements that organize LM's behavior with systematic procedures [14, 44, 6, 43] or symbolic program guidance. For example, Schlag et al. [27] embeds LMs in an algorithmic search procedure to help solve problems like question answering step-by-step, in which the search trees are expanded by relevant paragraphs that might provide answers. This approach however differs from ours in that trees are expanded by sampling external paragraphs instead of the LM's own thoughts, and there is no reflection or voting steps. Another approach, LLM+P [18], goes one step further and delegates the actual planning process to a classical planner.

**Classical search methods.** Last but not least, our approach can be treated as a modern rendition of classical search methods for problem solving. For example it can be considered as a heuristic search algorithm like A* [10], in which the heuristic at each search node is provided by the LM's self-assessment. From this perspective, our method is also related to NeuroLogic A*esque decoding [19], which is inspired by A* search but introduces look-ahead heuristics that are efficient for LMs to improve the beam-search or top-k sampling decoding. This method however is constrained to sentence generation tasks, whereas our framework are designed for complex, multi-step problem solving guarded by value feedback.

# 6    Discussion

**Limitations and future directions.** Deliberate search such as ToT might not be necessary for many existing tasks that GPT-4 already excels at (see Appendix B.1), and as an initial step this work only explores three relatively simple tasks that challenges GPT-4 (see Appendix B.2 for some GPT-3.5 experiment results) and calls of better search and planning abilities incorporated with LMs. However, as we begin to deploy LMs for more real-world decision making applications (e.g. coding, data analysis, robotics, etc.), more complex tasks could emerge and present new opportunities to study these research questions. Also, search methods like ToT requires more resources (e.g. GPT-4 API cost) than sampling methods in order to improve task performances, but the modular flexibility of ToT allows users to customize such performance-cost tradeoffs, and ongoing open-source efforts [32] should readily reduce such costs in the near future. More details about cost and efficiency are in Appendix B.3. Lastly, this work focuses on using an off-the-shelf LM, and fine-tuning LMs using a ToT-style high-level counterfactual decision making (e.g. deliberating over potential choices for the next paragraph, instead of predicting the next token) might present opportunities to enhance the problem-solving capabilities of LMs.

**Conclusion.** The associative "System 1" of LMs can be beneficially augmented by a "System 2" based on searching a tree of possible paths to the solution to a problem. The Tree of Thoughts framework provides a way to translate classical insights about problem-solving into actionable methods for contemporary LMs. At the same time, LMs address a weakness of these classical methods, providing a way to solve complex problems that are not easily formalized, such as creative writing. We see this intersection of LMs with classical approaches to AI as an exciting direction.

**Broader Impact**

ToT is a framework that empowers LMs to more autonomously and intelligently make decisions and solve problems. While current tasks are limited to reasoning and search problems, future applications involving interaction with external environments or humans could bring potential danger, e.g. facilitating harmful uses of LMs. On the other hand, ToT also improves the interpretability of model decisions and the opportunity for human alignment, as the resulting representations are readable, high-level language reasoning instead of implicit, low-level token values.

**Acknowledgements**

SY and KN acknowledge support from an Oracle Collaborative Research award and the National Science Foundation under Grant No. 2239363. Any opinions, findings, conclusions, or recommendations expressed in this material are those of the author(s) and do not necessarily reflect the views of the National Science Foundation. SY is also supported by the Harold W. Dodds Fellowship from Princeton.

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

# A Code, Prompts, Trajectories

All code is available at `https://github.com/princeton-nlp/tree-of-thought-llm`.

All prompts are available at `https://github.com/princeton-nlp/tree-of-thought-llm/tree/master/src/tot/prompts`.

Trajectories are available at `https://github.com/princeton-nlp/tree-of-thought-llm/tree/master/logs`.

# B Additional Experiment Results

Given the motivation of exploring and extending the capability frontier of language models, our experiments in the main paper have focused on a setup with the state-of-the-art language model (GPT-4), and three hard tasks invented to challenge it. Here, we report additional experiments with weaker LLM or easier tasks, and discuss cost and efficiency.

| | GSM8K | StrategyQA | | GPT-4 | GPT-3.5 | | GPT-4 | GPT-3.5 |
|-----|-------|------------|-----|-------|---------|-----|-------|---------|
| IO | 51 | 73 | IO | 7.3% | 6% | IO | 6.19 | 4.47 |
| CoT | 86 | 82 | CoT | 4.0% | 3% | CoT | 6.93 | 5.16 |
| ToT | **90** | **83** | ToT | **74%** | **19%** | ToT | **7.56** | **6.62** |

Table 4: New tasks with zero-shot ToT and GPT-4.

Table 5: Game of 24 with GPT-4 vs GPT-3.5.

Table 6: Creative Writing with GPT-4 vs. GPT-3.5.

## B.1 Extension to new tasks (GSM8k, StrategyQA) with zero-shot ToT

While more common NLP tasks might be too easy for GPT-4 and do not require ToT (which is why we considered harder new tasks), we believe applying ToT to new tasks could be straightforward. For example, we implemented a simple and generic zero-shot ToT-BFS similar to creative writing (sample 5 problem solving strategies then vote for the best one; then sample 5 solutions based on the best strategy then vote for the best one) for GSM8K and StrategyQA with few extra lines of code:

```
# define the answer format of new tasks
gsm8k_format = '"the answer is n" where n is a number'
strategyqa_format = 'either "the answer is yes" or "the answer is no"'

# define zero-shot io prompting
standard_prompt = 'Answer the following question with {format}: {input}'

# define thought format for zero-shot cot and zero-shot tot
cot_prompt = '''Answer the following question: {input}

Make a strategy then write. Your output should be of the following format:

Strategy:
Your strategy about how to answer the question.

Answer:
Your answer to the question. It should end with {format}.
'''

# define zero-shot voting used for zero-shot tot
vote_prompt = '''Given an instruction and several choices,
decide which choice is most promising.
Analyze each choice in detail, then conclude in the last line
"The best choice is {s}", where s the integer id of the choice.
'''
```

We evaluated on a subset of 100 random GSM8K test and StrategyQA dev questions. As shown in Table 4 and as expected, ToT improves over CoT on both tasks (but only slightly, given GPT-4 + CoT is already very good on such tasks, and StrategyQA's bottleneck is external knowledge, not reasoning). Considering computational costs, it is more suitable to try smaller LLMs + ToT for traditional NLP tasks, or GPT-4 + ToT for hard tasks that challenge GPT-4 + CoT's reasoning.

## B.2 Extension to new LMs (GPT-3.5)

To understand how ToT works with other LLMs, we also ran GPT-3.5-turbo for Creative Writing (Table 6) and Game of 24 (Table 5). On both tasks, "ToT > CoT > IO" remains true for GPT-3.5. On Creative Writing, we find GPT-3.5+ToT outperform GPT-4+IO, and similar to GPT-4+CoT, which suggests ToT could also work well on weaker language models.

On Game of 24 (we changed 1-shot proposal prompt to 3-shot to make it work), GPT-3.5+ToT's 19% is far worse than GPT-4+ToT's 74%. To further understand the importance of generation vs. evaluation, we ran GPT-4 generation + GPT-3.5 evaluation (64%) and GPT-3.5 generation + GPT-4 evaluation (31%). This suggests the game's bottleneck is thought generation, and different generation/evaluation language models might attain decent results while reducing costs.

## B.3 Cost and efficiency

Running ToT requires significantly more computations than IO or CoT prompting. For example, in Game of 24 (Table 7 below), solving a problem with ToT requires 5.5k completion tokens, close to 100 CoT trials (6.7k tokens). But the performance of ToT is better than best of 100 independent CoT trials.

| Game of 24 | Generate/Prompt tokens | Cost per case | Success |
|---|---|---|---|
| IO (best of 100) | 1.8k / 1.0k | $0.13 | 33% |
| CoT (best of 100) | 6.7k / 2.2k | $0.47 | 49% |
| ToT | 5.5k / 1.4k | $0.74 | 74% |

Table 7: Cost analysis on Game of 24.

On Creative Writing (Table 8 below), we found ToT takes around 5x completion tokens and money cost, which is intuitive as $b = 5$ and most tokens are generated passages.

| Creative Writing | Generate/Prompt tokens | Cost per case |
|---|---|---|
| IO | 0.9k / 0.4k | $0.06 |
| CoT | 0.9k / 0.4k | $0.07 |
| ToT | 4k / 2.9k | $0.32 |

Table 8: Cost analysis on Game of 24.

So completing Game of 24 and Creative Writing's main ToT experiments cost around $0.74 \times 100 + 0.32 \times 100 = 106$ dollars. Crosswords' DFS experiments should be also within 100 dollars. In general, cost and efficiency of ToT highly depend on the prompts and search algorithms used, and could require 5-100 times more generated tokens than CoT. Some actionable insights:

- We recommend using ToT on tasks requiring deliberate reasoning, on which CoT struggles.
- Flexibility of ToT allows some performance-cost tradeoff, e.g., change beam size or vote number in BFS, few-shot vs. zero-shot prompting, GPT-3.5 vs. GPT-4, etc. One could configure the setup based on some resource constraints or performance goal.
- There is much space for improving efficiency, e.g., BFS could early stop when solution is found, or trim down beam size to when some thoughts are "impossible".
- We believe that more computation is indeed required in order for the model to achieve stronger intelligence, and this should not become a blocking issue as in the long run, (open-source) LMs will become much cheaper and more efficient. It is also a great direction how to better train/finetune LMs for thought generation and/or evaluation.

