# OpenReview forum: "Tree of Thoughts: Deliberate Problem Solving with Large Language Models"
_NeurIPS.cc/2023/Conference — NeurIPS 2023 oral_

### Official Review · Reviewer_sEEP · 2023-07-02

**Soundness:** 3 good
**Presentation:** 3 good
**Contribution:** 2 fair
**Rating:** 6
**Confidence:** 4

**Summary:**

The paper introduces an innovative concept of problem-solving that utilizes a tree-like structure of thoughts, constructed and evaluated by the LLM, specifically GPT-4. To illustrate the efficacy of this technique, the authors have incorporated three distinct tasks: the Game of 24, creative writing, and crossword puzzles. These tasks have been chosen as they require different capabilities to be solved. Several iterations of the tree-like structure of thoughts were examined, including those combined with Breadth-First Search (BFS) or Depth-First Search (DFS), along with other variations with different hyperparameters. A comprehensive comparison has been drawn with various other Language Learning Model-based baselines.

**Strengths:**

The value of examining the tree-like thinking process performed by LLM is evident to the scientific community. Primarily, the Tree of Thoughts (ToT) could be applicable to other tasks when adequately adapted and it provides an intriguing exploration of GPT-4's capabilities.

The paper intriguingly selects tasks requiring diverse skill sets:
- The game of 24 demands fundamental mathematical capabilities and the ability to assess whether success is achievable from a partial solution.
- Creative writing needs the generation of coherent and sound text under strict constraints.
- Crossword puzzles necessitate linguistic knowledge and the capacity to search across a vast state space (large word sets that meet given constraints).

The authors have made a commendable effort to objectively assess the results, with blind tests of text coherency serving as a prime example. The elucidation of the method is lucid, with Figure 1 being particularly illuminating. Additionally, most experiments are presented in a comprehensible manner. A significant strength lies in ToT's performance, which significantly surpasses that of the baseline.

I perceive this paper to be a successful proof-of-concept for ToT. Notably, its significance chiefly stems from the ongoing massive efforts aimed at leveraging LLM's capabilities. A prominent challenge in applying LLMs for deliberate problem solving

**Weaknesses:**

The biggest weakness of ToT is that the paper lacks the evaluation of some important properties of ToT. Here I put a list:

1. There is no data about the price of ToT compared with baselines (there is a short mention in the Limitations about the price/success rate trade-off). In the best case, I would like to have a graph where there is a success rate on the y-axis and a price on the x-axis. If you think that such a graphical analysis would be too detailed please put some data in a table. The price of GPT-4 may be different in the future, so alternatively you can add information about the number of tokens used by ToT per solution compared with baselines. Probably the best would be to have both information: price and number of tokens.

2. There is no data about the time execution of ToT compared with baselines.

Comment on 1. and 2. : Knowing the average price and time is very important for readers who consider building their own methods on top of ToT or consider applying ToT to their task.

3. Some additional comparisons are missing. We know that a single execution of ToT is significantly more powerful than a single execution of baseline (for example IO). However is it possible that running IO or CoT many times (until it finds a solution) is actually cheaper and faster than ToT? ToT will still have scientific value even in the case where the answer to the previous question is positive, but the scientific community will benefit from knowing it. Currently, it is not really known what are the maximum capabilities of properly used LLM and how to maximize its performance. In a search for the most effective methods, we need to know as many properties of each approach as possible. Knowing more about ToT would add great value to the paper. Here is a list of my suggestions for additional comparisons:
> How many times on average should I run a baseline to find a solution compared with ToT?
> Can baseline be cheaper or faster than ToT even though it is weaker?
If it is impossible to gather detailed data for this comparison, please provide some estimates.

4. While I see value in the creative writing task, I am not very convinced that a task with just one ToT step is meaningful when evaluating tree search algorithms. More precisely: while formally this task falls under the ToT framework, it's hard to think about it in terms of tree search.


**Questions:**

Necessary things (either to address here or in the paper):
- Figure 3b:  The title of the figure is “Samples failed at each step”, and there last two bars show success not failures (the one named “Correct”). It is misleading, I first thought that ToT failed mainly at the last step. Another problem with this comparison is that ToT with b=5 vs CoT seems unfair to me (ToT has many samples at each state while CoT does not - it is not surprising that CoT stands no chance). The better comparison would be for example: ToT vs  CoT-SC  or ToT vs CoT-best of 100.
- In Table 2 and Figure 3 there is no information about the error estimation. It should be at least in text somewhere
The total cost of the experiments done during the work on this paper should be stated in the paper or in supplementary materials.
- line 186. I would change "left" to "remaining" (numbers). For a while, I was wondering if left refers to "left to do" or "left side".
- line 134: dot missing at the end of the sentence
- Algorithm 2 and line 279: how exactly does the pruning work?
- lines 222 to 225: Here you describe an iterative-refine method as a baseline. Would such a method be reasonable for baselines in Game of 24?

Suggestions, ideas, and comments:
-All the tasks presented here could be solved without ToT (and two of them even without LLM).  Maybe you have some intuition for what kind of problems ToT would be necessary? I know that this may be hard or even impossible to answer.
- It would be very interesting to see a comparison of ToT based on GPT3.5 vs IO GPT4 or CoT GPT4. Can weaker LLM with ToT be stronger than Sota LLM? I think such an experiment would add great value to the paper.
- The comment at the bottom of page 8 is very interesting





**Limitations:**

The limitations section is sound.

---

> ### Author Rebuttal · Authors · 2023-08-09
>
>
> Thank you for your detailed and constructive feedback!
>
> ### 1. Cost and efficiency
> This is a great point. Please see **General Response (3)**.
>
> ### 2. Running IO/CoT baselines many times
>
> We showed Game of 24 IO/CoT best-of-k results in Table 2 and Figure 3, where CoT best-of-100 has a game success of 49%, which is still much worse than ToT's 74%. We also showed iterative refinement approaches for Game of 24 (Table 2) and Creative Writing (Figure 5). These findings suggest ToT might be a better way to spend more resources in hope to get better results for these tasks, compared with parallel sampling or iterative refinement.
>
>
> ### 3. Creative Writing Steps
>
> We note that the ToT for creative writing has two steps: plans are generated and evaluated, then passages are generated and evaluated based on the best plan.
>
> ### 4. What tasks need ToT
>
> This is a great question! Please see **General Response (1)**, where we show how easy it is to adapt ToT to more tasks. We think GPT-4+ToT is more suitable for hard tasks challenging GPT-4+CoT, while weaker LLMs+ToT can be used for simpler tasks.
>
> ### 5. ToT with weaker LLMs
>
> Please see **General Response (2)**.
>
> ### 6. Other smaller questions
>
> - Figure 3b: thanks for the suggestion, we will remove last two "success" bars. The figure is not intended to contrast the performances of ToT over CoT, but rather to support the point that CoT mostly fails at the first step of Game of 24 due to the inherent problems of autoregressive decoding, thus exploration around initial decisions is crucial.
>
> - Table 2 and Figure 3: we did not include error estimation as the performance gaps are significant and running GPT-4 experiments are expensive. As detailed in **General Response (3)**, running ToT experiments on Game of 24 and Creative Writing cost around 100 dollars.
>
> - We will fix line 186 and 134, thanks for your careful reading!
>
> - DFS pruning: as stated in line 265-266, a state is pruned if LLM deems any remaining clue as "impossible" to fill in (e.g. To heap: tm_s_).
>
> - Iterative refine for Game of 24: We tried it, and as shown in Table 2's IO + Refine (k=10, which is already very expensive due to the culcumative context), it does help a bit, but the performance of 27% is still poor compared to ToT's 74%.

---

> > ### Comment · Reviewer_sEEP · 2023-08-14
> > **Thanks for the response**
> >
> > Thanks for providing the additional data. The most interesting and promising is that GPT 3.5 with ToT can outperform GPT4 IO.

---

### Official Review · Reviewer_3c3c · 2023-07-04

**Soundness:** 3 good
**Presentation:** 3 good
**Contribution:** 3 good
**Rating:** 8
**Confidence:** 4

**Summary:**

This paper proposes Tree of Thoughts to promote deliberate problem solving with LLMs. By using a tree-based structure and a four-step process towards problem solving, tree of thoughts successfully address many of the challenges with left-to-right decoding such as looking ahead, backtracking, considering multiple reasoning paths at the same time, and more. Extensive experiments on three reasoning tasks demonstrate the superiority of Tree of Thoughts over ICL, CoT, self consistency, and more.

**Strengths:**

+ the Tree of Thoughts approach is well motivated by the limitations of autoregressive decoding
+ the four-step formulation is novel and clearly described
+ experiments on three reasoning tasks are extensive and convincing

**Weaknesses:**

Overall I like this work and I only have a few comments.

- I wonder if it might be possible to discuss the efforts required to adapt Tree of Thoughts to other tasks and reasoning formats. To me, it seems like other than the search algorithm (BFS/DFS), the other three parts would require some hand-crafted engineering from scratch for a different task.

- I wonder if least to most [1] or decomposed prompting [2] might be possible baselines for the three tasks: they also discuss some sort of iterative problem solving potential.

- In the "Game of 24" task, does the base LLM make arithmetic mistakes? If that's the case, would using tool learning strategies [3] further improve Tree of Thoughts in numeric reasoning tasks?

- The supplementary material does not seem to contain an appendix, while some details might be helpful. For example, what is the prompt for GPT-4 zero-shot evaluation in line 213? What are the details of human evaluation for "Creative Writing"? (annotator head count, compensation, etc.) Also, it is increasingly common to present all the prompt texts in the appendix to facilitate reproducibility. I wonder if the authors might consider having an appendix to systematically document all details about the approach and experiments.

[1] Zhou, Denny, et al. "Least-to-Most Prompting Enables Complex Reasoning in Large Language Models." The Eleventh International Conference on Learning Representations. 2022.

[2] Khot, Tushar, et al. "Decomposed Prompting: A Modular Approach for Solving Complex Tasks." The Eleventh International Conference on Learning Representations. 2022.

[3] Schick, Timo, et al. "Toolformer: Language models can teach themselves to use tools." arXiv preprint arXiv:2302.04761 (2023).

**Questions:**

please see above

**Limitations:**

Limitations are discussed in Section 6.

---

> ### Author Rebuttal · Authors · 2023-08-09
>
>
> Thank you for finding our work "well-motivated", "novel", "clearly described", and "convincing".
>
> ### 1. Adapt ToT to other tasks.
>
> This is a great question. Please see **General Response (1)**, where we show a simple scheme that adapts ToT to StrategyQA and GSM8K with near-zero task-specific hand crafting.
>
> ### 2. Least-to-most or decomposed prompting
>
> We think they might not be effective baslines for our studied tasks, where initial decisions (e.g. first step of game of 24, or first filled word in crosswords) are critical and require exploration. Methods like least-to-most or decomposed prompting might help with compositional generalization via task decomposition, but do not have a way to explore and maintain different decomposition plans. So once the first step is generated wrong, they might fail similarly as CoT.
>
> ### 3. Arithmetic mistake and tool use
>
> In Game of 24, numbers are usually within 50, and GPT-4 rarely makes arithmetic mistakes (weaker models like GPT-3.5 make more mistakes).
>
> As hinted in Page 8's footnote, we agree it is a great idea and important future direction to enhence ToT with external tool use! We probably need some better and harder tasks that require exploration of both reasoning and acting.
>
> ### 4. Appendix
>
> Thank you for the reminder! We will include additioanl experiment details and all prompts used in the appendix.

---

> > ### Comment · Reviewer_3c3c · 2023-08-15
> >
> > Thank you for the response. I have increased my rating and wish the authors good luck.

---

### Official Review · Reviewer_fAU5 · 2023-07-07

**Soundness:** 3 good
**Presentation:** 3 good
**Contribution:** 2 fair
**Rating:** 5
**Confidence:** 4

**Summary:**

The paper presents a new framework for language model inference called Tree of Thoughts, which aims to improve the ability of language models to solve complex, multi-step problem-solving tasks.

The framework involves generating a tree of possible plans for solving a given task, with each node in the tree representing a possible step in the plan. The language model then evaluates each node based on its own self-assessment, and the tree is expanded by sampling external paragraphs to generate new nodes. The framework also includes a voting step to determine the best plan to follow.

The paper argues that this approach is more effective than previous methods for prompting language models, such as Chain of Thought, and can be seen as a modern rendition of classical search methods for problem-solving. The contributions of the paper include a detailed description of the Tree of Thoughts framework, an evaluation of its effectiveness on a range of problem-solving tasks, and a discussion of its potential limitations and future directions for research.


**Strengths:**

1. The idea of upgrading chain-of-thoughts into tree-of-thoughts seems to be an intuitive extension and necessary step. There are many benefits of tree reasoning, including the ability to look ahead and backtrack to search for better traces.
2. The formulation of heuristics leverages the recent self-evaluation ability of GPT-4 to reuse the LLM to evaluate the quality of states via prompting. This relieves the necessity of training accurate state values and enables approximate estimation.
3. The empirical results on three tasks demonstrate the effectiveness of ToT compared with GPT-4 based baselines. The improvement gain is quite large on two tasks, considering GPT-4 based CoT methods don't work well on them.

**Weaknesses:**

1. Unsurprisingly, the idea of extending linear reasoning to tree reasoning isn't a new thing. Specifically, [1] leverages beam search to guide the chain-of-thought to decode a better reasoning path, which is almost the same as the BFS in this paper. [1] also uses self-evaluation to provide heuristics. [2] also formulates the reasoning as a tree reasoning problem. More importantly, the application of BFS and DFS seems to be ad-hoc, and there could be more principled methods to guide the search. For example, [3, 4] applies MCTS to guide the search, which might have better planning abilities.
2. While ToT improves CoT naturally, it doesn't come without cost. One major concern is the efficiency issue for querying the expensive GPT-4 multiple times. The paper should give the audience a clearer idea of how cost ToT is compared with CoT, probably with an efficiency comparison between them. More interestingly, the paper should propose or at least discuss potential means to reduce the inference cost, such as using various smaller models for sampling or state evaluation, etc.
3. The application scope is largely limited. While the authors say they select the three tasks because they are hard, the selected three tasks are arguably narrow with two text games. One possible reason is that the selected tasks make the thought and state formulation easier and demonstrate improvement significantly. Nonetheless, it would be necessary to extend the scope to be more similar to CoT tasks to demonstrate the broader applicability of the proposed methods.

[1]. Xie, Yuxi, et al. "Decomposition enhances reasoning via self-evaluation guided decoding." arXiv preprint arXiv:2305.00633 (2023).
[2]. Jung, Jaehun, et al. "Maieutic prompting: Logically consistent reasoning with recursive explanations." arXiv preprint arXiv:2205.11822 (2022).
[3]. Zhu, Xinyu, et al. "Solving math word problem via cooperative reasoning induced language models." arXiv preprint arXiv:2210.16257 (2022).
[4]. Hao, Shibo, et al. "Reasoning with language model is planning with world model." arXiv preprint arXiv:2305.14992 (2023).


**Questions:**

I raised multiple questions in the weakness section, and I hope the authors could further clarify them. Specifically, some can be reformualted as follows:
1. How hard is it to apply the proposed method to a more general NLP reasoning dataset, like GSM8K or StrategyQA, or other well-known datasets? If possible, I hope the authors could demonstrate one case using the proposed method to solve a more general task.
2. How necessary the GPT-4 is in the proposed framework? It seems GPT-4 is very powerful for self-evaluation, and how critical is this component for the success of the tasks?

---

> ### Author Rebuttal · Authors · 2023-08-09
>
>
> Thank you for your thoughtful comments, all of which are very helpful for improving our work!
>
> ### 1. Related work and BFS/DFS vs. MCTS
> Thank you for pointing out these recent or concurrent papers related to ToT. We will discuss them in our related work section.
>
> We used BFS and DFS as they are the simplest tree search algorithms that turn out general and effective enough for the studied tasks. Due to the modularity of the ToT framework, application of more advanced algorithms such as MCTS or A* for harder tasks is a clear and promising future direction.
>
> ### 2. Cost and potential means to more efficiency
> These are great points. Please see **General Response (3)**.
>
> ### 3. Extend the scope to more CoT tasks
> Please see **General Response (1)**, where as per your suggestion, we show a very simple scheme to extend ToT to StrategyQA and GSM8K.
>
> ### 4. Importance of GPT-4
> Please see **General Response (2)**.

---

> > ### Comment · Reviewer_fAU5 · 2023-08-19
> > **Thanks for the rebuttal**
> >
> > I appreciate the response and the new experiment results. My score remains the same for the following reasons:
> > 1. Though one pair of experiment comparison shows that gpt-3.5+ToT outperforms gpr4+IO, in most cases, gpt-3.5+ToT doesn't work, esp, when used as the generation model. Therefore, whether the proposed work will be generalized to other models, esp., open-sourced models, is questionable or unanswered for me. If the proposed method can only work when using GPT-4, its cost will be one of the major obstacles for people to use it, also demonstrated by the new cost-efficiency result.
> > 2. This paper is a well-polished one with near ideas implemented very well and I have no doubt about its influence on the community in the short near future. However, its core novelty of applying (simple) search algorithms to LLM reasoning isn't significant enough.

---

> > > ### Author Response · Authors · 2023-08-21
> > > **Thanks for reply**
> > >
> > > Thanks for your reply!
> > >
> > > 1. In both tasks, we've shown GPT-3.5+ToT can outperform GPT-4+CoT, and the proposed method is not specific to GPT-4. We believe it is very possible to better prompt LLMs (e.g. GPT-3.5) or finetune LLMs (e.g. Llama) to achieve better ToT performances in the near future (e.g. while one-shot proposal prompt in game of 24 was good enough for GPT-4, changing it to three-shot helped a lot for GPT-3.5; more prompt tuning can be done to yield low-hanging fruits).
> > > 2. Thanks for endorsing ToT's "influence on the community in the short near future"! Like you said, current (open-source) LLMs might be limited in their capabilities in ToT-style generations and evaluations; also, we don't have enough hard tasks that challenge GPT-4+CoT's deliberate reasoning. Improving new models and proposing new tasks are long-term community efforts beyond a single paper, and that's exactly why we believe ToT's influence could be long-term, by pointing out what generation and evaluation capabilities LLMs should improve on, what kind of tasks should be devised to challenge LLMs, and in general, linking frontier LLM research to classic and everlasting insights at the root of AI and CogSci.

---

### Official Review · Reviewer_ysyd · 2023-07-08

**Soundness:** 4 excellent
**Presentation:** 4 excellent
**Contribution:** 4 excellent
**Rating:** 8
**Confidence:** 4

**Summary:**

This paper introduces a new method for prompting large language models (LLMs) for multi-step reasoning tasks. Existing prompting methods are confined to the autoregressive generation scheme, making it difficult for LLMs to finish tasks that require exploration and planning. To alleviate this problem, the authors propose Tree of Thought (ToT), which combines the chain-of-thought (CoT) method with tree-based search.  By prompting LLMs to solve the intermediate step and using the self-evaluations as the heuristics, one can effectively leverage LLMs to finish tasks that require exploration and backtracking. Empirical evaluations on three novel tasks demonstrate the effectiveness of the proposed method.

**Strengths:**

- A novel method to combine LLMs with tree-based search. The proposed method alleviates the shortcoming of directly prompting LLMs to generate answers for some tasks that require explorations and backtracking. Compared to CoT prompting which can only sample one solution path, ToT allows LLMs to explore more potential solutions and find the best one.
- Automatic and independent reasoning process. ToT fully depends on the LLMs to generate plans, finish intermediate steps, and generate self-evaluation for the current state as heuristics. Without the dependence on external tools or models, ToT enables LLMs to automatically finish complex tasks.

**Weaknesses:**

More application scenarios. The authors mainly evaluate ToT on the three novel tasks that require exploration and backtracking. It would be better to demonstrate whether ToT can help improve LLMs on commonly-used reasoning tasks.

**Questions:**

Please see the weakness above.

**Limitations:**

The authors discussed the limitations and potential negative social impact of their work

---

> ### Author Rebuttal · Authors · 2023-08-09
>
> Thank you for endorsing our work!
>
> ### 1. More application scenarios
>
> This is a great point. Please check **General Response (1)**, where we show a very simple scheme to apply ToT in common NLP tasks (StrategyQA, GSM8K). However, such tasks might not need GPT-4 + ToT as GPT-4 + COT suffices --- weaker LLMs + ToT could potentially outperform weaker LLMs + CoT and could be studied on such tasks.

---

> > ### Comment · Reviewer_ysyd · 2023-08-21
> >
> > I have carefully read the authors' responses. I would like to keep my rating. Just a quick comment for more potential ablation studies. One way to adopt LLMs to solve the Game of 24 is to ask LLMs to write a Python program -- in that case, we leave the exploration and search tasks to the program rather than LLM itself. The Python program will finish the search for the final solution through BFS or DFS. That could be a very strong baseline for ToT. This can make the comparison with CoT more comprehensive and fair -- CoT is weaker at search and exploration, but it can write programs to finish the search process.

---

### Author Rebuttal · Authors · 2023-08-09

We appreciate all reviewer's great feedback, which will significantly strengthen our draft!

The motivation of ToT is simple: **to explore and extend the capability frontier of autoregressive LLMs**. More specifically, given the SoTA LLM (GPT-4) can already solve many existing NLP tasks, what new tasks can raise new challenges? How can we augment LLMs to tackle these challenges? To this end, we contribute three new tasks for LLMs that challenge even GPT-4, and ToT as a framework to augment LLM's deliberate reasoning. **Our paper is thus focused on a setup with SoTA LLM (GPT-4) and hard tasks for it.**

Here, we report new experiments with weaker LLM or easier tasks, and discuss cost and efficiency concerns.

### 1. New Experiments on Other (Easier) Tasks (ysyd, fAU5, 3c3c)

While more common NLP tasks might be too easy for GPT-4 and do not require ToT (which is why we considered harder new tasks), we believe **applying ToT to new tasks could be straightforward**.
For example, we implemented a simple and generic zero-shot ToT-BFS similar to creative writing (sample 5 problem solving strategies then vote for the best one; then sample 5 solutions based on the best strategy then vote for the best one) for GSM8K and StrategyQA with few extra lines of code:

```python
gsm8k_format = '"the answer is n" where n is a number'

strategyqa_format = 'either "the answer is yes" or "the answer is no"'

standard_prompt = 'Answer the following question with {format}: {input}'

cot_prompt = '''
Answer the following question: {input}

Make a strategy then write. Your output should be of the following format:

Strategy:
Your strategy about how to answer the question.

Answer:
Your answer to the question. It should end with {format}.
'''

vote_prompt = '''Given an instruction and several choices, decide which choice is most promising. Analyze each choice in detail, then conclude in the last line "The best choice is {s}", where s the integer id of the choice.
'''
```

We evaluated on a subset of 100 random GSM8K test and StrategyQA dev questions. As shown below and as expected, ToT improves over CoT on both tasks (but only slightly, given GPT-4 + CoT is already very good on such tasks, and StrategyQA's bottleneck is external knowledge, not reasoning). Considering computational costs, it is more suitable to try smaller LLMs + ToT for traditional NLP tasks, or GPT-4 + ToT for hard tasks that challenge GPT-4 + CoT's reasoning.

|     | GSM8k  | StrategyQA |
| -------- | ------- |------- |
| IO  | 51    | 73    |
| CoT | 86     |  82   |
| ToT | 90    |  83    |

### 2. New Experiments on Other (Weaker) LLM (fAU5, sEEP)

To understand how ToT works with other LLMs, we also ran GPT-3.5-turbo for creative writing, which was reported in the supplementary material. **We find gpt-3.5+ToT outperform gpt-4+IO, and similar to gpt-4+CoT, which suggests ToT could also work well on weaker LLM.**

|Creative Writing| gpt-4 (in paper) | gpt-3.5 |
|-----|-----|----|
| IO | 6.19| 4.47|
| CoT | 6.93| 5.16|
| ToT | 7.56| 6.62|

We also ran GPT-3.5 for Game of 24 (we changed 1-shot proposal prompt to 3-shot to make it work). Here, GPT-3.5+ToT's 19% is far worse than GPT-4+ToT's 74%.

|  Game of 24  | gpt-4 (in paper) | gpt-3.5 |
| ---- | ---- |---- |
| IO  | 7.3    | 6    |
| CoT | 4.0     | 3     |
| ToT |  74  |  19    |

To further understand the importance of generation vs. evaluation, we ran gpt-4 gen + gpt-3.5 eval (64%) and gpt-3.5 gen + gpt-4 eval (31%). This suggests the game's bottleneck is thought generation, and different gen/eval LLMs might attain decent results while reducing costs.

|  gen\eval | gpt-4 | gpt-3.5 |
| ---- | ---- |---- |
| gpt-4  | 74    | 64    |
| gpt-3.5 | 31     | 19     |

### 3. Cost and efficiency (fAU5, sEEP)

Running ToT requires significantly more computations than IO or CoT prompting. For example, in game of 24, solving a problem with ToT requires 5.5k completion tokens, close to 100 CoT trials (6.7k tokens). But the performance of ToT is better than best of 100 CoT trials.

|  Game of 24 | completion tokens | prompt tokens | cost per case | success rate |
| ---- | ---- |---- | ---- |  ---- |
| IO (best of 100) | 1.8k    | 1.0k   | $0.13 | 33% |
| CoT (best of 100) | 6.7k     | 2.2k     | $0.47 | 49%
| ToT | 5.5k    |  1.4k    | $0.74| 74% |

On Creative Writing, we found ToT takes around 5x completion tokens and money cost, which is intuitive as $b=5$ and most tokens are generated passages.

| Creative Writing   | completion tokens | prompt tokens | cost per case |
| ---- | ---- |---- | ---- |
| IO  | 0.9k    | 0.4k   | $0.06 |  |
| CoT | 0.9k     | 0.4k     | $0.07 |
| ToT | 4k    |  2.9k   | $0.32 |

So completing Game of 24 and Creative Writing's main ToT experiments cost around $0.74 * 100 + 0.32 * 100 = 106$ dollars. Unfortunately we did not record the usage for crosswords' DFS experiments, but it should be within 100 dollars. In general, cost and efficiency of ToT highly depend on the prompts and search algorithms used, and could require 5-100 times more generated tokens than CoT.

Some actionable insights:

- We recommend using ToT on complex tasks requiring deliberate reasoning, on which CoT struggles.
- Flexibility of ToT allows some performance-cost tradeoff, e.g. change beam size or vote number in BFS, few-shot vs. zero-shot prompting, GPT-3.5 vs. GPT-4, etc. One could configure the setup based on some resource constraints or performance goal.
- There is much space for improving efficiency --- e.g. BFS could early stop when solution is found, or trim down beam size to $b<5$ when some thoughts are "impossible".
- We believe that more computation is indeed required in order for the model to achieve stronger intelligence, and this should not become a blocking issue as in the long run, (open-source) LLMs will become much cheaper and more efficient. It is also a great direction how to better train/finetune LLMs for thought generation and/or evaluation.

---

### Author Response · Authors · 2023-08-17
**Thanks for responses and look forward to discussions**

We want to thank Reviewers 3c3c and sEEP for responding to our rebuttal, and we look forward to engaging in discussions with Reviewers ysyd and fAU5 and addressing any potential remaining concerns in the remaining 4 days. Thanks in advance!

---

### Decision · Program_Chairs · 2023-09-21

**Decision:**

Accept (oral)

**Comment:**

I will recommend this paper for acceptance, because reviewers felt the paper proposed an interesting approach (ysyd, fAU5, 3c3c, sEEP), and showed strong improvements over CoT (fAU5).

There was extensive conversation between reviewers and authors in the rebuttal period. Weak points were raised by reviewers and were predominantly addressed by authors in rebuttal, including (i) more tasks would be nice (ysyd, fAU5, 3c3c) to which rebuttal added more applications (gSM8K and StrategyQA), (ii) efficiency issues (fAU5), (iii) parts of the work aren’t interesting/novel enough (fAU5; “I have no doubt about its influence on the community in the short near future. However, its core novelty of applying (simple) search algorithms to LLM reasoning isn't significant enough.“), (iv) necessity of GPT4 in the pipeline (fAU5, sEEP), which was also addressed in rebuttal.

Additional baselines were suggested (3c3c, sEEP), but during reviewer and AC discussion, reviewers verified that these were optional, and that the paper meets the bar in their opinions regardless.

Recommendations for the next version of the work: We recommend summarizing discussion from rebuttal in the paper, including regarding the tradeoffs of ToT v. CoT (sEEP) including efficiency trade off of CoT (fAU5), the difficulty of implementing the approach (3c3c, sEEP). The prompt details suggested by 3c3c should also be included in the next version of the paper. It would also be nice to see the results with earlier LMs+ToT included in the final paper.